# A multi-agent reinforcement learning model of common-pool resource appropriation

**Julien Perolat**\*
DeepMind
London, UK
perolat@google.com

**Joel Z. Leibo**\*
DeepMind
London, UK
jzl@google.com

**Vinicius Zambaldi**
DeepMind
London, UK
vzambaldi@google.com

**Charles Beattie**
DeepMind
London, UK
cbeattie@google.com

**Karl Tuyls**
University of Liverpool
Liverpool, UK
karltuyls@google.com

**Thore Graepel**
DeepMind
London, UK
thore@google.com

## Abstract

Humanity faces numerous problems of common-pool resource appropriation. This class of multi-agent social dilemma includes the problems of ensuring sustainable use of fresh water, common fisheries, grazing pastures, and irrigation systems. Abstract models of common-pool resource appropriation based on non-cooperative game theory predict that self-interested agents will generally fail to find socially positive equilibria—a phenomenon called the tragedy of the commons. However, in reality, human societies are sometimes able to discover and implement stable cooperative solutions. Decades of behavioral game theory research have sought to uncover aspects of human behavior that make this possible. Most of that work was based on laboratory experiments where participants only make a single choice: how much to appropriate. Recognizing the importance of spatial and temporal resource dynamics, a recent trend has been toward experiments in more complex real-time video game-like environments. However, standard methods of non-cooperative game theory can no longer be used to generate predictions for this case. Here we show that deep reinforcement learning can be used instead. To that end, we study the emergent behavior of groups of independently learning agents in a partially observed Markov game modeling common-pool resource appropriation. Our experiments highlight the importance of trial-and-error learning in common-pool resource appropriation and shed light on the relationship between exclusion, sustainability, and inequality.

## 1 Introduction

Natural resources like fisheries, groundwater basins, and grazing pastures, as well as technological resources like irrigation systems and access to geosynchronous orbit are all common-pool resources (CPRs). It is difficult or impossible for agents to exclude one another from accessing them. But whenever an agent obtains an individual benefit from such a resource, the remaining amount available for appropriation by others is ever-so-slightly diminished. These two seemingly-innocent properties of CPRs combine to yield numerous subtle problems of motivation in organizing collective action [12, 26, 27, 6]. The necessity of organizing groups of humans for effective CPR appropriation, combined with its notorious difficulty, has shaped human history. It remains equally critical today.

Renewable natural resources[†] have a stock component and a flow component [10, 35, 7, 26]. Agents may choose to appropriate resources from the flow. However, the magnitude of the flow depends on the state of the stock[‡]. Over-appropriation negatively impacts the stock, and thus has a negative impact on future flow. Agents secure individual rewards when they appropriate resource units from a CPR. However, the cost of such appropriation, felt via its impact on the CPR stock, affects all agents in the community equally. Economic theory predicts that as long as each individual's share of the marginal social cost is less than their marginal gain from appropriating an additional resource unit, agents will continue to appropriate from the CPR. If such over-appropriation continues unchecked for too long then the CPR stock may become depleted, thus cutting off future resource flows. Even if an especially clever agent were to realize the trap, they still could not unilaterally alter the outcome by restraining their own behavior. In other words, CPR appropriation problems have socially-deficient Nash equilibria. In fact, the choice to appropriate is typically dominant over the choice to show restraint (e.g. [32]). No matter what the state of the CPR stock, agents prefer to appropriate additional resources for themselves over the option of showing restraint, since in that case they receive no individual benefit but still endure the cost of CPR exploitation by others.

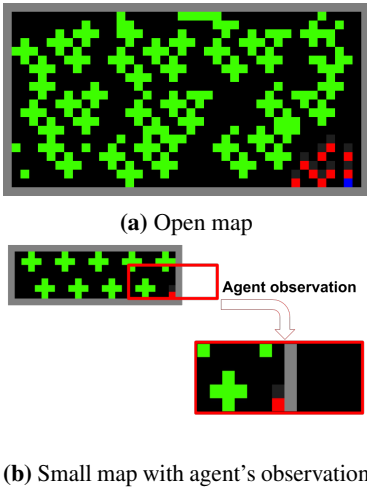

**(a)** Open map

**(b)** Small map with agent's observation

**Figure 1:** **(a)** The initial state of the Commons Game at the start of each episode on the large open map used in sections 3.2, 3.3, and 3.5. Apples are green, walls are grey, and players are red or blue. **(b)** The initial state of the small map used for the single-agent experiment (Section 3.1). The size of the window of pixels a player receives as an observation is also shown.

Nevertheless, despite such pessimistic theoretical predictions, human communities frequently are able to self-organize to solve CPR appropriation problems [26, 28, 27, 6]. A major goal of laboratory-based behavioral work in this area is to determine what it is about human behavior that makes this possible. Being based on behavioral game theory [4], most experimental work on human CPR appropriation behavior features highly abstracted environments where the only decision to make is how much to appropriate (e.g. [29]). The advantage of such a setup is that the theoretical predictions of non-cooperative game theory are clear. However, this is achieved by sacrificing the opportunity to model spatial and temporal dynamics which are important in real-world CPRs [26]. This approach also downplays the role of trial-and-error learning.

One recent line of behavioral research on CPR appropriation features significantly more complex environments than the abstract matrix games that came before [16, 18, 17, 14, 15]. In a typical experiment, a participant controls the movements of an on-screen avatar in a real-time video game-like environment that approximates a CPR with complex spatial and temporal dynamics. They are compensated proportionally to the amount of resources they collect. Interesting behavioral results have been obtained with this setup. For example, [18] found that participants often found cooperative solutions that relied on dividing the CPR into separate territories. However, due to the increased complexity of the environment model used in this new generation of experiments, the standard tools of non-cooperative game theory can no longer be used to generate predictions.

We propose a new model of common-pool resource appropriation in which learning takes the center stage. It consists of two components: (1) a spatially and temporally dynamic CPR environment, similar to [17], and (2) a multi-agent system consisting of $N$ independent self-interested deep reinforcement learning agents. On the collective level, the idea is that self-organization to solve CPR appropriation problems works by smoothly adjusting over time the incentives felt by individual agents through a process akin to trial and error. This collective adjustment process is the aggregate result of all the many individual agents simultaneously learning how best to respond to their current situation.

---

[†]Natural resources may or may not be renewable. However, this paper is only concerned with those that are.

[‡]CPR appropriation problems are concerned with the allocation of the flow. In contrast, CPR *provision* problems concern the supply of the stock. This paper only addresses the appropriation problem and we will say no more about CPR provision. See [7, 26] for more on the distinction between the two problems.

This model of CPR appropriation admits a diverse range of emergent social outcomes. Much of the present paper is devoted to developing methodology for analyzing such emergence. For instance, we show how behavior of groups may be characterized along four *social outcome metrics* called: efficiency, equality, sustainability, and peace. We also develop an $N$-player empirical game-theoretic analysis that allows one to connect our model back to standard non-cooperative game theory. It allows one to determine classical game-theoretic properties like Nash equilibria for strategic games that emerge from learning in our model.

Our point is not to argue that we have a more realistic model than standard non-cooperative game theory. This is also a reductionist model. However, it emphasizes different aspects of real-world CPR problems. It makes different assumptions and thus may be expected to produce new insights for the general theory of CPR appropriation that were missed by the existing literature's focus on standard game theory models. Our results are broadly compatible with previous theory while also raising a new possibility, that trial-and-error learning may be a powerful mechanism for promoting sustainable use of the commons.

## 2 Modeling and analysis methods

### 2.1 The commons game

The goal of the Commons Game is to collect "apples" (resources). The catch is that the apple regrowth rate (i.e. CPR flow) depends on the spatial configuration of the uncollected apples (i.e the CPR stock): the more nearby apples, the higher the regrowth rate. If all apples in a local area are harvested then none ever grow back—until the end of the episode (1000 steps), at which point the game resets to an initial state. The dilemma is as follows. The interests of the individual lead toward harvesting as rapidly as possible. However, the interests of the group as a whole are advanced when individuals refrain from doing so, especially in situations where many agents simultaneously harvest in the same local region. Such situations are precarious because the more harvesting agents there are, the greater the chance of bringing the local stock down to zero, at which point it cannot recover.

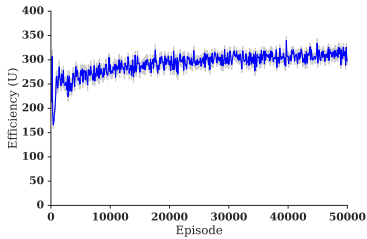

**(a)** Single agent return

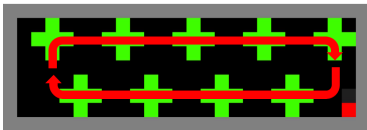

**(b)** Optimal path

**Figure 2: (a)** Single-agent returns as a function of training steps. **(b)** The optimal resource appropriation policy for a single agent on this map. At convergence, the agent we study nearly learns this policy: https://youtu.be/NnghJgsMxAY.

So far, the proposed Commons Game is quite similar to the dynamic game used in human behavioral experiments [16, 18, 17, 14, 15]. However, it departs in one notable way. In the behavioral work, especially [17], participants were given the option of paying a fee in order to fine another participant, reducing their score. In contrast, in our Commons Game, agents can tag one another with a "time-out beam". Any agent caught in the path of the beam is removed from the game for 25 steps. Neither the tagging nor the tagged agent receive any direct reward or punishment from this. However, the tagged agent loses the chance to collect apples during its time-out period and the tagging agent loses a bit of time chasing and aiming, thus paying the opportunity cost of foregone apple consumption. We argue that such a mechanism is more realistic because it has an effect within the game itself, not just on the scores.

The Commons Game is a partially-observable general-sum Markov Game [33, 22]. In each state of the game, agents take actions based on a partial observation of the state space and receive an individual reward. Agents must learn through experience an appropriate behavior policy while interacting with one another.

In technical terms, we consider an $N$-player partially observable Markov game $\mathcal{M}$ defined on a finite set of states $\mathcal{S}$. The observation function $O : \mathcal{S} \times \{1, \ldots, N\} \to \mathbb{R}^d$ specifies each player's $d$-dimensional view on the state space. In any state, players are allowed to take actions from the set $\mathcal{A}^1, \ldots, \mathcal{A}^N$ (one for each player). As a result of their joint action $a^1, \ldots, a^N \in \mathcal{A}^1, \ldots, \mathcal{A}^N$ the state changes following the stochastic transition function $\mathcal{T} : \mathcal{S} \times \mathcal{A}^1 \times \cdots \times \mathcal{A}^N \to \Delta(\mathcal{S})$ (where $\Delta(\mathcal{S})$ denotes the set of discrete probability distributions over $\mathcal{S}$) and every player receives an individual reward defined as

$r^i : \mathcal{S} \times \mathcal{A}^1 \times \cdots \times \mathcal{A}^N \to \mathbb{R}$ for player $i$. Finally, let us write $\mathcal{O}^i = \{o^i \mid s \in \mathcal{S}, o^i = O(s,i)\}$ be the observation space of player $i$.

Each agent learns, independently through their own experience of the environment, a behavior policy $\pi^i : \mathcal{O}^i \to \Delta(\mathcal{A}^i)$ (written $\pi(a^i|o^i)$) based on their own observation $o^i = O(s,i)$ and reward $r^i(s, a^1, \ldots, a^N)$. For the sake of simplicity we will write $\vec{a} = (a^1, \ldots, a^N)$, $\vec{o} = (o^1, \ldots, o^N)$ and $\vec{\pi}(.|\vec{o}) = (\pi^1(.|o^1), \ldots, \pi^N(.|o^N))$. Each agent's goal is to maximize a long term $\gamma$-discounted payoff defined as follow:

$$V_{\vec{\pi}}^i(s_0) = \mathbb{E}\left[\sum_{t=0}^{\infty} \gamma^t r^i(s_t, \vec{a}_t) | \vec{a}_t \sim \vec{\pi}_t, s_{t+1} \sim \mathcal{T}(s_t, \vec{a}_t)\right]$$

## 2.2 Deep multi-agent reinforcement learning

Multi-agent learning in Markov games is the subject of a large literature [3], mostly concerned with the aim of *prescribing* an optimal learning rule. To that end, many algorithms have been proposed over the past decade to provide guarantees of convergence in specific settings. Some of them address the zero-sum two-player case [22], or attempt to solve the general-sum case [13, 11]. Others study the emergence of cooperation in partially observable Markov decision processes [9, 37, 38] but rely on knowledge of the model which is unrealistic when studying independent interaction. Our goal, as opposed to the *prescriptive* agenda, is to describe the behaviour that emerges when agents learn in the presence of other learning agents. This agenda is called the *descriptive* agenda in the categorization of Shoham & al. [34]. To that end, we simulated $N$ independent agents, each simultaneously learning via the deep reinforcement learning algorithm of Mnih et al. (2015) [24].

Reinforcement learning algorithms learn a policy through experience balancing exploration of the environment and exploitation. These algorithms were developed for the single agent case and are applied independently here [21, 3] even though this multi-agent context breaks the Markov assumption [20]. The algorithm we use is $Q$-learning with function approximation (i.e. DQN) [24]. In $Q$-learning, the policy of agent $i$ is implicitly represented through a state-action value function $Q^i(O(s,i), a)$ (also written $Q^i(s,a)$ in the following). The policy of agent $i$ is an $\epsilon$-greedy policy and is defined by $\pi^i(a|O(s,i)) = (1 - \epsilon)\mathbf{1}_{a=\arg\max_a Q^i(s,a)} + \frac{\epsilon}{|\bar{A}^i|}$. The parameter $\epsilon$ controls the amount of exploration. The $Q$-function $Q^i$ is learned to minimize the bellman residual $\|Q^i(o^i, a^i) - r^i - \max_b Q^i(o'^i, b)\|$ on data collected through interaction with the environment $(o^i, a^i, r^i, o'^i)$ in $\{(o_t^i, a_t^i, r_t^i, o_{t+1}^i)\}$ (where $o_t^i = O(s_t, i)$).

## 2.3 Social outcome metrics

Unlike in single-agent reinforcement learning where the value function is the canonical metric of agent performance, in multi-agent systems with mixed incentives like the Commons Game, there is no scalar metric that can adequately track the state of the system (see e.g. [5]). Thus we introduce four key social outcome metrics in order to summarize group behavior and facilitate its analysis.

Consider $N$ independent agents. Let $\{r_t^i \mid t = 1, \ldots, T\}$ be the sequence of rewards obtained by the $i$-th agent over an episode of duration $T$. Likewise, let $\{o_t^i \mid t = 1, \ldots T\}$ be the $i$-th agent's observation sequence. Its return is given by $R^i = \sum_{t=1}^T r_t^i$.

The *Utilitarian metric* ($U$), also known as *Efficiency*, measures the sum total of all rewards obtained by all agents. It is defined as the average over players of sum of rewards $R^i$. The *Equality* metric ($E$) is defined using the Gini coefficient [8]. The *Sustainability* metric ($S$) is defined as the average time at which the rewards are collected. The *Peace* metric ($P$) is defined as the average number of untagged agent steps.

$$U = \mathbb{E}\left[\frac{\sum_{i=1}^N R^i}{T}\right], \quad E = 1 - \frac{\sum_{i=1}^N \sum_{j=1}^N |R^i - R^j|}{2N \sum_{i=1}^N R^i}, \quad S = \mathbb{E}\left[\frac{1}{N}\sum_{i=1}^N t^i\right] \quad \text{where } t^i = \mathbb{E}[t \mid r_t^i > 0].$$

$$P = \frac{\mathbb{E}\left[NT - \sum_{i=1}^N \sum_{t=1}^T I(o_t^i)\right]}{T} \quad \text{where } I(o) = \begin{cases} 1 & \text{if } o = \text{time-out observation} \\ 0 & \text{otherwise.} \end{cases}$$

# 3   Results

## 3.1   Sustainable appropriation in the single-agent case

In principle, even a single agent, on its own, may learn a strategy that over-exploits and depletes its own private resources. However, in the single-agent case, such a strategy could always be improved by individually adopting a more sustainable strategy. We find that, in practice, agents are indeed able to learn an efficient and sustainable appropriation policy in the single-agent case (Fig. 2).

## 3.2   Emergent social outcomes

Now we consider the multi-agent case. Unlike in the single agent case where learning steadily improved returns (Fig. 2-a), in the multi-agent case, learning does not necessarily increase returns. The returns of a single agent are also a poor indicator of the group's behavior. Thus we monitor how the social outcome metrics that we defined in Section 2.3 evolve over the course of training (Fig. 3).

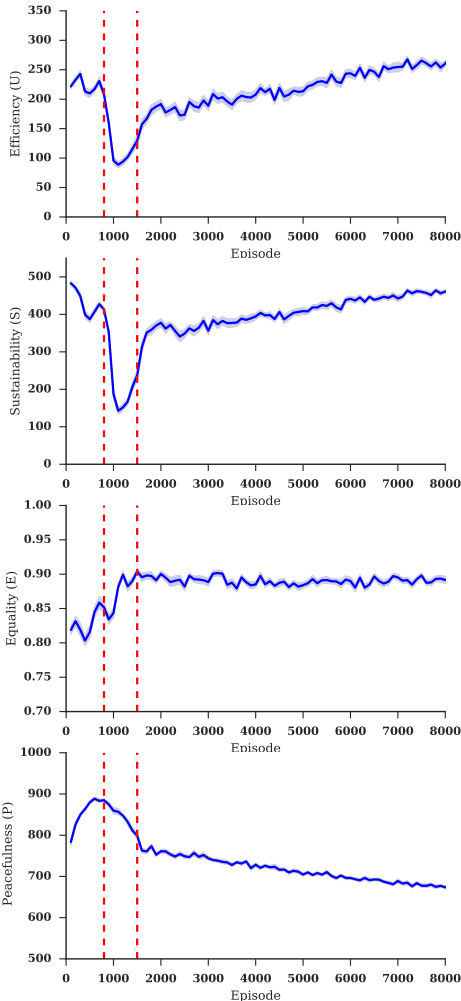

**Figure 3:** Evolution of the different social outcome metrics (Sec.2.3) over the course of training on the open map (Fig.1a) using a time-out beam of length 10 and width 5. From top to bottom is displayed, the utility metric $(U)$, the sustainability metric $(S)$, the equality metric $(E)$, and the peace metric $(P)$.

The system moves through 3 phases characterized by qualitatively different behaviors and social outcomes. Phase 1, which we may call *naïvety*, begins at the start of training and extends until $\approx 900$ episodes. It is characterized by healthy CPR stocks (high apple density). Agents begin training by acting randomly, diffusing through the space and collecting apples whenever they happen upon them. Apples density is high enough that the overall utilitarian efficiency $(U)$ is quite high, and in fact is close to the max it will ever attain. As training progresses, agents learn to move toward regions of greater apple density in order to more efficiently harvest rewards. They detect no benefit from their tagging action and quickly learn not to use it. This can be seen as a steady increase in the peace metric $(P)$ (Fig. 3). In a video[§] of typical agent behavior in the naïvety phase, it can be seen that apples remain plentiful (the CPR stock remains healthy) throughout the entire episode.

Phase 2, which we may call *tragedy*, begins where naïvety ends ($\approx$ episode 900), it is characterized by rapid and catastrophic depletion of CPR stock in each episode. The sustainability metric $(S)$, which had already been decreasing steadily with learning in the previous phase, now takes a sudden and drastic turn downward. It happens because agents have learned "too well" how to appropriate from the CPR. With each agent harvesting as quickly as they possibly can, no time is allowed for the CPR stock to recover. It quickly becomes depleted. As a result, utilitarian efficiency $(U)$ declines precipitously. At the low point, agents are collecting less than half as many apples per episode as they did at the very start of training—when they were acting randomly (Fig. 3). In a video[¶] of agent play at the height of the tragedy one can see that by $\approx 500$ steps into the (1100-step) episode, the stock has been completely depleted and no more apples can grow.

Phase 3, which we may call *maturity*, begins when efficiency and sustainability turn the corner and start to recover again after their low point ($\approx$ episode 1500) and continues indefinitely. Initially, conflict breaks out when agents discover that, in situations of great

---

[§]learned policy after 100 episodes `https://youtu.be/ranlu_9ooDw`.
[¶]learned policy after 1100 episodes `https://youtu.be/1xF1DoLxqyQ`.

apple scarcity, it is possible to tag another agent to prevent them from taking apples that one could otherwise take for themselves. As learning continues, this conflict expands in scope. Agents learn to tag one another in situations of greater and greater abundance. The peace metric ($P$) steadily declines (Fig. 3). At the same time, efficiency ($U$) and sustainability ($S$) increase, eventually reaching and slightly surpassing their original level from before tragedy struck. How can efficiency and sustainability increase while peace declines? When an agent is tagged by another agent's beam, it gets removed from the game for 25 steps. Conflict between agents in the Commons Game has the effect of lowering the effective population size and thus relieving pressure on the CPR stock. With less agents harvesting at any given time, the survivors are free to collect with greater impunity and less risk of resource depletion. This effect is evident in a video[‖] of agent play during the maturity phase. Note that the CPR stock is maintained through the entire episode. By contrast, in an analogous experiment with the tagging action disabled, the learned policies were much less sustainable (Supp. Fig. 11).

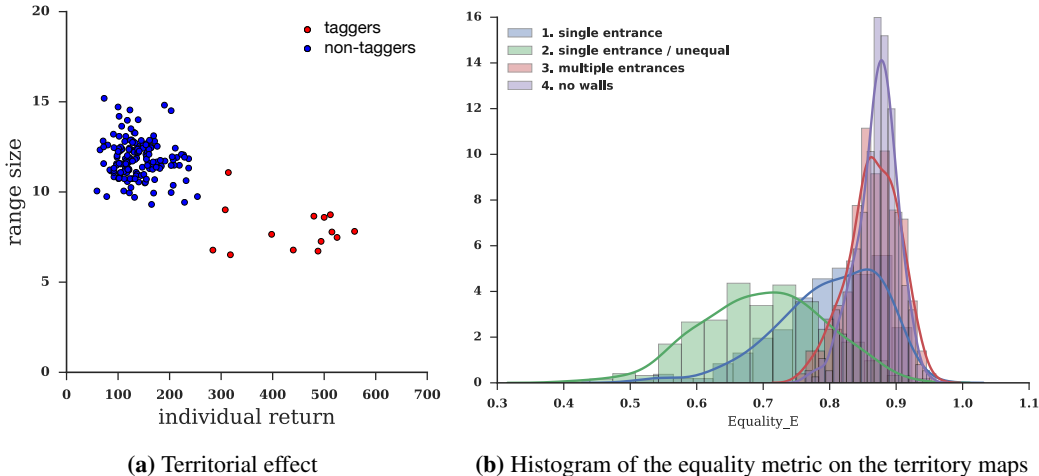

**(a)** Territorial effect           **(b)** Histogram of the equality metric on the territory maps

**Figure 4: (a)** Scatter plot of return by range size (variance of position) for individual agents in experiments with one tagging agent (red dots, one per random seed) and 11 non-tagging agents (blue dots, eleven per random seed). The tagging players collect more apples per episode than the others and remain in a smaller part of the map. This illustrates that the tagging players take over a territory and harvest sustainably within its boundary. **(b)** represents the distribution of the equality metric ($E$) for different runs on four different maps with natural regions from which it may be possible to exclude other. The first map is the standard map from which others will be derived (Fig. 6c). In the second apples are more concentrated on the top left corner and will respawn faster (Fig. 6d). the third is porous meaning it is harder for an agent to protect an area (Fig 6e). On the fourth map, the interiors walls are removed (Fig 6f). Figure 4b shows inequality rises in maps where players can exclude one another from accessing the commons.

### 3.3 Sustainability and the emergence of exclusion

Suppose, by building a fence around the resource or some other means, access to it can be made exclusive to just one agent. Then that agent is called the owner and the resource is called a private good [30]. The owner is incentivized to avoid over-appropriation so as to safeguard the value of future resource flows from which they and they alone will profit. In accord with this, we showed above (Fig. 2) that sustainability can indeed be achieved in the single agent case. Next, we wanted to see if such a strategy could emerge in the multi-agent case.

The key requirement is for agents to somehow be able to exclude one another from accessing part of the CPR, i.e. a region of the map. To give an agent the chance to exclude others we had to provide it with an advantage. Thus we ran an experiment where only one out of the twelve agents could use the tagging action. In this experiment, the tagging agent learned a policy of controlling a specific territory by using its time-out beam to exclude other agents from accessing it. The tagging agents roam over a smaller part of the map than the non-tagging agents but achieve better returns (Fig. 4a). This is because the non-tagging agents generally failed to organize a sustainable appropriation pattern

---

[‖] learned policy after 3900 episodes `https://youtu.be/XZXJYgPuzEI`.

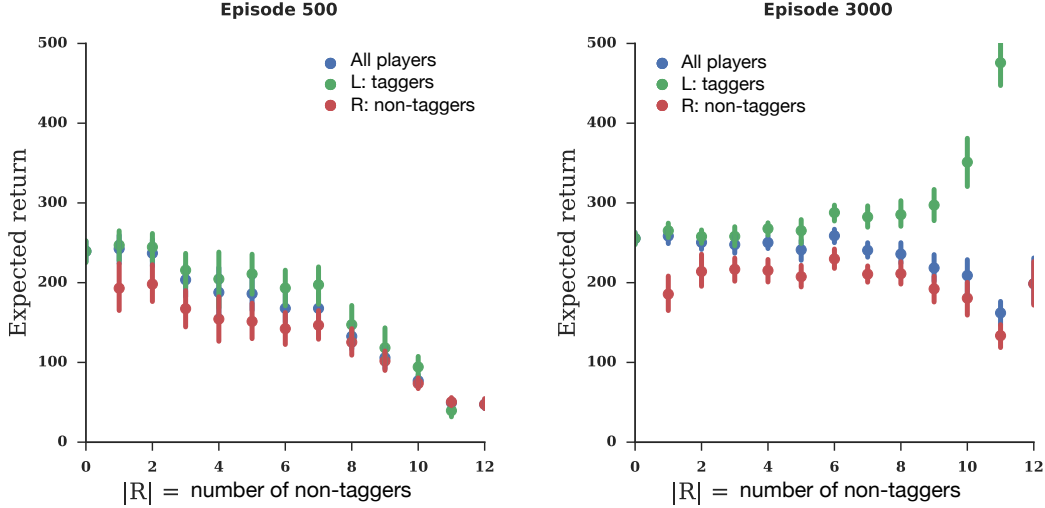

**(a)** Early training (after 500 episodes) Schelling diagram for $L$ = taggers and $R$ = non-taggers

**(b)** Late training (after 3,000 episodes) Schelling diagram for $L$ = taggers and $R$ = non-taggers

**Figure 5:** Schelling diagram from early (5a) and late (5b) in training for the experiment where $L$ = taggers and $R$ = non-taggers.

and depleted the CPR stock in the area available to them (the majority of the map). The tagging agent, on the other hand, was generally able to maintain a healthy stock within its "privatized" territory**.

Interestingly, territorial solutions to CPR appropriation problems have emerged in real-world CPR problems, especially fisheries [23, 1, 36]. Territories have also emerged spontaneously in laboratory experiments with a spatially and temporally dynamic commons game similar to the one we study here [18].

### 3.4 Emergence of inequality

To further investigate the emergence of exclusion strategies using agents that all have the same abilities (all can tag), we created four new maps with natural regions enclosed by walls (see Supp. Fig. 6). The idea is that it is much easier to exclude others from accessing a territory that has only a single entrance than one with multiple entrances or one with no walls at all. This manipulation had a large effect on the equality of outcomes. Easier exclusion led to greater inequality (Fig. 4b). The lucky agent that was first to learn how to exclude others from "its territory" could then monopolize the lion's share of the rewards for a long time (Supp. Figs. 7a and 7b). In one map with unequal apple density between the four regions, the other agents were never able to catch up and achieve returns comparable to the first-to-learn agent (Supp. Fig. 7b). On the other hand, on the maps where exclusion was more difficult, there was no such advantage to being the first to learn (Supp. Figs. 7c and 7d).

### 3.5 Empirical game-theoretic analysis of emergent strategic incentives

We use empirical game theoretic analysis to characterize the strategic incentives facing agents at different points over the course of training. As in [21], we use simulation to estimate the payoffs of an abstracted game in which agents choose their entire policy as a single decision with two alternatives. However, the method of [21] cannot be applied directly to the case of $N > 2$ players that we study in this paper. Instead, we look at Schelling diagrams [32]. They provide an intuitive way to summarize the strategic structure of a symmetric $N$-player 2-action game where everyone's payoffs depend only on the number of others choosing one way or the other. Following Schelling's terminology, we refer to the two alternatives as $L$ and $R$ (left and right). We include in the appendix several examples of Schelling diagrams produced from experiments using different ways of assigning policies to $L$ and $R$ groups (Supp. Fig. 8).

---

**A typical episode where the tagging agent has a policy of excluding others from a region in the lower left corner of the map: `https://youtu.be/3iGnpijQ8RM`.

In this section we restrict our attention to an experiment where $L$ is the choice of adopting a policy that uses the tagging action and $R$ the choice of a policy that does not tag. A Schelling diagram is interpreted as follows. The green curve is the average return obtained by a player choosing $L$ (a tagger) as a function of the number of players choosing $R$ (non-taggers). Likewise, the red curve is the average return obtained by a player choosing $R$ as a function of the number of other players also choosing $R$. The average return of all players is shown in blue. At the leftmost point, $|R| = 0 \implies |L| = N$, the blue curve must coincide with the green one. At the rightmost point, $|R| = N \implies |L| = 0$, the blue curve coincides with the red curve.

Properties of the strategic game can be read off from the Schelling diagram. For example, in Fig. 5b one can see that the choice of a tagging policy is dominant over the choice of a non-tagging policy since, for any $|R|$, the expected return of the $L$ group is always greater than that of the $R$ group. This implies that the Nash equilibrium is at $|R| = 0$ (all players tagging). The Schelling diagram also shows that the collective maximum (blue curve's max) occurs when $|R| = 7$. So the Nash equilibrium is socially-deficient in this case.

In addition to being able to describe the strategic game faced by agents at convergence, we can also investigate how the strategic incentives agents evolve over the course of learning. Fig. 5a shows that the strategic game after 500 training episodes is one with a *uniform* negative externality. That is, no matter whether one is a tagger or a non-tagger, the effect of switching one additional other agent from the tagging group to the non-tagging group is to decrease returns. After 3000 training episodes the strategic situation is different (Fig. 5b). Now, for $|R| > 5$, there is a *contingent* externality. Switching one additional agent from tagging to non-tagging has a positive effect on the remaining taggers and a negative effect on the non-taggers (green and red curves have differently signed slopes).

## 4 Discussion

This paper describes how algorithms arising from reinforcement learning research may be applied to build new kinds of models for phenomena drawn from the social sciences. As such, this paper really has two audiences. For social scientists, the core conclusions are as follows. (1) Unlike most game theory-based approaches where modelers typically "hand engineer" specific strategies like tit-for-tat [2] or win-stay-lose-shift [25], here agents must learn *how* to implement their strategic decisions. This means that the resulting behaviors are emergent. For example, in this case the tragedy of the commons was "solved" by reducing the effective population size below the environment's carrying capacity, but this outcome was not assumed. (2) This model endogenizes exclusion. That is, it allows agents to learn strategies wherein they exclude others from a portion of the CPR. Then, in accord with predictions from economics [26, 1, 18, 36], sustainable appropriation strategies emerge more readily in the "privatized" zones than they do elsewhere. (3) Inequality emerges when exclusion policies are easier to implement. In particular, natural boundaries in the environment make inequality more likely to arise.

From the perspective of reinforcement learning research, the most interesting aspect of this model is that—despite the fact that all agents learn only toward their individual objectives—tracking individual rewards over the course of training is insufficient to characterize the state of the system. These results illustrate how multiple simultaneously learning agents may continually improve in "competence" without improving their expected discounted returns. Indeed, learning may even decrease returns in cases where too-competent agents end up depleting the commons. Without the social outcome metrics (efficiency, equality, sustainability, and peace) and other analyses employed here, such emergent events could not have been detected. This insight is widely applicable to other general-sum Markov games with mixed incentives (e.g. [19, 21]).

This is a reductionist approach. Notice what is conspicuously absent from the model we have proposed. The process by which groups of humans self-organize to solve CPR problems is usually conceptualized as one of rational negotiation (e.g. [26]). People do things like bargain with one another, attempt to build consensus for collective decisions, think about each other's thoughts, and make arbitration appeals to local officials. The agents in our model can't do anything like that. Nevertheless, we still find it is sometimes possible for self-organization to resolve CPR appropriation problems. Moreover, examining the pattern of success and failure across variants of our model yields insights that appear readily applicable to understanding human CPR appropriation behavior. The question then is raised: how much of human cognitive sophistication is really needed to find adequate solutions to CPR appropriation problems? We note that nonhuman organisms also solve them [31]. This suggests that trial-and-error learning alone, without advanced cognitive capabilities, may sometimes be sufficient for effective CPR appropriation.

## Footnotes

\*indicates equal contribution

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
