[Supplementary Material]

# Appendix

## A Simulation methods

The partially-observed Markov game environment underlying our implementation of the Commons Game is a 2D grid-world game engine. It works as follows. The state $s_t$ and the joint action of all players $\vec{a}$ determine the state at the next time-step $s_{t+1}$. Observations $O(s, i) \in \mathbb{R}^{3 \times 20 \times 21}$ (RGB) of the true state $s_t$ depend on the $i$-th player's current position and orientation. The observation window extends 20 grid squares ahead and 10 grid squares from side to side (see Fig. 1). Actions $a \in \mathbb{R}^8$ were agent-centered: step forward, step backward, step left, step right, rotate left, rotate right, tag, and stand still. The beam extends 20 grid squares in the direction faced by the tagger. It is 5 squares wide. If another agent is in the path of the beam then it is tagged. A tagged agent is removed from the game for 25 timesteps. Being tagged has no direct impact on rewards. However, there is an indirect effect since a tagged agent cannot collect apples during its time-out period. Each player appears blue in its own local view and red in its opponent's view. Each episode lasted for $1,000$ steps. The local flow of apple respawning depends on local stock size. Specifically, the per-timestep respawn probability $p_t$ of a potential apple at position $c$ depends on the number of already-spawned apples in a ball of radius 2 centered around its location, i.e. the local stock size $L = |\{\text{already-spawned apples} \in B_2(c)\}|$. The per-timestep respawn probability as a function of local stock size is given by:

$$p_t(L) = \begin{cases} 0 & \text{if } L = 0 \\ 0.01 & \text{if } L = 1 \text{ or } 2 \\ 0.05 & \text{if } L = 3 \text{ or } 4 \\ 0.1 & \text{if } L > 4 \end{cases} \tag{1}$$

Default neural networks had two hidden layers with 32 units, interleaved with rectified linear layers which projected to the output layer which had 8 units, one for each action. During training, players implemented epsilon-greedy policies, with epsilon decaying linearly over time (from $1.0$ to $0.1$). The default per-time-step discount rate $\gamma$ was $0.99$.

**(a)** Small map      **(b)** Open map      **(c)** Basic single-entrance region map

**(d)** Unequal single-entrance region map      **(e)** Multi-entrance region map      **(f)** Region map with no walls

**Figure 6:** We used six different maps in our experiments. Map 6a is a small environment where a single random player will sometimes harvest all the apples. Map 6b is considerably larger. It was designed so that a single agent, acting alone cannot remove all the apples but with several agents harvesting simultaneously it becomes relatively easy to remove them all. Maps 6c, 6e, and 6f were constructed in order to manipulate the ease with which agents can exclude one another from accessing territories defined by natural boundaries (walls). They were designed to be compared to one another. Thus all three have the same number and spatial configuration of apples. Map 6c has only one single entrance to each region. Map 6e has two additional entrances to each region and in map 6f, all walls were removed. Map 6d was created to test the effects of territories with unequal productivity. In its top left corner region, the apples are placed closer to one another than in the other regions. Thus, since the growth rule is density dependent, apples respawn faster in this region (greater CPR flow).

**(a)** Basic single-entrance region map (6c)

**(b)** Unequal single-entrance region map (6d)

**(c)** Multi-entrance region map (6e)

**(d)** Region map with no walls (6f)

**Figure 7:** These figures shows how inequality emerges and persists on different maps. In this experiment, 12 players learns simultaneously on maps 6c, 6d, 6e, and 6f. In each plot, the red time series shows reward as a function of training time for the agent that was most successful in the first 1000 episodes (before the vertical red dashed line). The aim of this analysis was to determine whether an early advantage would persist over time. That is, was the inequality persistent, with the same agent coming out on top in episode after episode? Or was it more random, with different agents doing well in each episode? As expected from Fig. 4b, in the multi-entrance and no-walls maps, this analysis shows no inequality. Interestingly, the two cases where inequality emerged are different from one another. The advantage of the first-to-learn agent on the basic single-entrance map (6c) was transient. However on the unequal single-entrance map (6d), its advantage was persistent.

**(a)** Schelling diagram for fast and slow players where $L =$ fast and $R =$ slow

**(b)** Schelling diagram for players with small and large $\gamma$ where $L =$ hight $\gamma$ ($\gamma = 0.99$) and $R =$ low $\gamma$ ($\gamma = 0.9$)

**(c)** Schelling diagram for players with small and large networks where $L =$ small network (hidden layer $= (5)$) and $R =$ large network (hidden layer $= (32, 32)$)

**(d)** Schelling diagram for tagging and non-tagging players where $L =$ taggers and $R =$ non-taggers.

**Figure 8:** Schelling diagrams at convergence. Fig. 8a shows the interaction between fast and slow players, Fig. 8b shows the interaction between players with large and small $\gamma$ ($\gamma = 0.9$ or $0.99$), Fig. 8c shows the interaction between players with large and small neural network, and Fig. 8d shows the interaction between players that have the ability to tag or not.

**Figure 9:** Schelling diagrams during training for the experiment where $L$ = taggers and $R$ = non-taggers.

**Figure 10:** Schelling diagrams during training for the experiment where $L$ = fast and $R$ = slow.

**Figure 11:** Left: Evolution of the different social outcome metrics (Sec.2.3) over the course of training on the open map (Fig.1a) for the case without the time-out beam tagging action. Right: the same figure from the main text is reproduced for comparison. From top to bottom is displayed, the utilitarian efficiency metric ($U$), the sustainability metric ($S$), the equality metric ($E$), and the peace metric ($P$). Notice that in the case without the time-out beam, the final sustainability value is considerably lower than in the case with the time-out beam. This means that agents frequently deplete their resources before the end of the episode in the conflict-free case.