[Reviews · NeurIPS 2017]

Reviewer 1



An interesting take on modelling the CPR dilemma. From a social science perspective and purely as future work, it would be interesting to see the following: 1. How the agents behave if they are aware of the type of opponents in their local view (zapper/non-zapper). Does it lead them to form a (temporary?) alliance or do they tend to avoid each other. 2. The effects of varying the rate at which resources are harvested for a part of the agent population.

Reviewer 2



* Summary Authors introduce a novel multi-agent problem (from machine learning's perspective) of common-pool resource appropriation and study how independently learning agents, while trying to optimize certain incentives, can learn to avoid "tragedy of the commons". The authors find that in certain environments, the learned strategies are consistent with predictions from economics. * Strengths The paper is well written and argued paper introducing a novel problem applying techniques of multi-agent reinforcement learning to model possible mechanisms by which intelligent agents can learn to avoid "tragedy of the commons". It shows how modern machine learning methods, when applied appropriately can add more fidelity to the model under study, while still maintaining some tractability. The authors clearly mention the limitations of the model and avoid making any comments about the social-economical implications. * Weaknesses - The paper is not self contained Understandable given the NIPS format, but the supplementary is necessary to understand large parts of the main paper and allow reproducibility. I also hereby request the authors to release the source code of their experiments to allow reproduction of their results. - Use of deep-reinforcement learning is not well motivated The problem domain seems simple enough that a linear approximation would have likely sufficed? The network is fairly small and isn't "deep" either. - > We argue that such a mechanism is more realistic because it has an effect within the game itself, not just on the scores This is probably the most unclear part. It's not clear to me why the paper considers one to be more realistic than the other rather than just modeling different incentives? Probably not enough space in the paper but actual comparison of learning dynamics when the opportunity costs are modeled as penalties instead. As economists say: incentives matter. However, if the intention was to explicitly avoid such explicit incentives, as they _would_ affect the model-free reinforcement learning algorithm, then those reasons should be clearly stated. - Unclear whether bringing connections to human cognition makes sense As the authors themselves state that the problem is fairly reductionist and does not allow for mechanisms like bargaining and negotiation that humans use, it's unclear what the authors mean by Perhaps the interaction between cognitively basic adaptation mechanisms and the structure of the CPR itself has more of an effect on whether self-organization will fail or succeed than previously appreciated.'' It would be fairly surprising if any behavioral economist trying to study this problem would ignore either of these things and needs more citation for comparison against "previously appreciated". * Minor comments ** Line 16: > [18] found them... Consider using \citeauthor{} ? ** Line 167: > be the N -th agent’s should be i-th agent? ** Figure 3: Clarify what the fill implies and how many runs were the results averaged over? ** Figure 4: Is not self contained and refers to Fig. 6 which is in the supplementary. The figure is understandably large and hard to fit in the main paper, but at least consider clarifying that it's in the supplementary (as you have clarified for other figures from the supplementary mentioned in the main paper). ** Figure 5: - Consider increasing the axes margins? Markers at 0 and 12 are cut off. - Increase space between the main caption and sub-caption. ** Line 299: From Fig 5b, it's not clear that |R|=7 is the maximum. To my eyes, 6 seems higher.

Reviewer 3



General comment: The paper is well structured, the research well carried and was pleasant to read. The contribution of the paper is made clear and lies in two aspects: - Development of a game like environment to model the CPR appropriation model. - Description of a learned policy by multiple deep reinforcement learning agents. The authors make the descriptive goal very clear and I think the description work is rigorous and well achieved. The authors are trying to answer a relevant problem of CPR appropriation. The goal is to understand and describe possible emerging behavior in common-pool resource appropriation. The environment: On the first part, I must admit that my background in economy and social sciences is not enough to gage if the environment is relevant to model the CPR problem. However, the authors cite many sources from the field and claim that their model is more realistic. Some quantitative results to compare the new POMDP environment to previous work should be added. If it is not the main concern of the author it should be at least in the supplementary material. The description of the POMDP environment is well explained, the supplementary material is necessary to understand the environment, especially the observation space. I think the caption of Figure 1 should be changed to make more obvious the difference between states and observations. One of the most interesting aspect of the paper is the definition of the social-outcome metrics (section 2.3). I think these are very interesting and strongly help the interpretation of the resulting policies. To the best of my knowledge, drawing conclusions on the behavior of the agents based on these metrics is a novel approach and a good contribution. The results: The conclusions drawn are corrects and reasonable. A lot of quantitative graphs are present to justify the conclusions. I found the videos extremely helpful in understanding the behaviors. They designed an interesting set of experiments that allowed the emergence of non-trivial behaviors. In a RL point of view I think this is a key aspect of the work. The analysis of the emerging behaviors from the learning curve is well carried thanks to the relevant metrics. The authors compare their results with previous work qualitatively and make rather clear the advantages and limitations of their approach. I am not sure if it is possible but having a table or a graph with quantitative metrics would have been even more assertive. Conclusions: Regarding the social science aspect, the authors qualitatively compare their work to the existing literature. In my opinion, a quantitative result would have helped making the contribution to the social science domain clearer. The authors are aware of the limitations of the model and raise relevant open questions that could be addressed as future work. On the reinforcement learning aspect, they propose a novel approach to describe multi-agent behaviors and use novel metrics that help both the description work and the design of new experiments. I think this approach could be extended to other domains as future work. Minor comments: - A link to the code should have been added to the supplementary material for reproducibility of the results. - Figure 2 could be closer to the paragraph referring to it (Section 3) - The explanation of the Schelling diagram could have been clearer. - Rename discussion section to conclusion?